# Factorized linear discriminant analysis for phenotype-guided representation learning of neuronal gene expression data

## Abstract

A central goal in neurobiology is to relate the expression of genes to the structural and functional properties of neuronal types, collectively called their phenotypes. Single-cell RNA sequencing can measure the expression of thousands of genes in thousands of neurons. How to interpret the data in the context of neuronal phenotypes? We propose a supervised learning approach that factorizes the gene expression data into components corresponding to individual phenotypic characteristics and their interactions. This new method, which we call factorized linear discriminant analysis (FLDA), seeks a linear transformation of gene expressions that varies highly with only one phenotypic factor and minimally with the others. We further leverage our approach with a sparsity-based regularization algorithm, which selects a few genes important to a specific phenotypic feature or feature combination. We applied this approach to a single-cell RNA-Seq dataset of Drosophila T4/T5 neurons, focusing on their dendritic and axonal phenotypes. The analysis confirms results obtained by conventional methods but also points to new genes related to the phenotypes and an intriguing hierarchy in the genetic organization of these cells.

## 1 Introduction

The complexity of neural circuits is a result of many different types of neurons that specifically connect to each other. Each neuronal type has its own phenotypic traits, which together determine the role of the neuronal type in a neural circuit. Typical phenotypic descriptions of neurons include features such as dendritic and axonal laminations, electrophysiological properties, and connectivity (Sanes & Masland, 2015; Zeng & Sanes, 2017; Gouwens et al., 2019). However, the genetic programs behind these phenotypic characteristics are still poorly understood.

Recent progress in characterizing neuronal cell types and investigating their gene expression, especially with advances in high-throughput single-cell RNA-Seq (Zeng & Sanes, 2017), provides an opportunity to address this challenge. With massive data generated from single-cell RNA-Seq, we now face a computational problem: how to factorize the high-dimensional data into gene expression modules that are meaningful to neuronal phenotypes? Specifically, given phenotypic descriptions of neuronal types, such as their dendritic stratification and axonal termination, can one project the original data into a low-dimensional space corresponding to these phenotypic features and their interactions, and further extract genes critical to each of these components?

Here we propose a new analysis method named factorized linear discriminant analysis (FLDA). Inspired by multi-way analysis of variance (ANOVA) (Fisher, 1918), this method factorizes data into components corresponding to phenotypic features and their interactions, and seeks a linear transformation that varies highly with one specific factor but not with the others. The linear nature of this approach makes it easy to interpret, as the weight coefficients directly inform the relative importance of each gene to each factor. We further introduce a sparse variant of the method, which constrains the number of genes contributing to each linear projection. We illustrate this approach by applying FLDA to a single-cell transcriptome dataset of T4/T5 neurons in Drosophila (Kurmangaliyev et al., 2019), focusing on two phenotypes: dendritic location and axonal lamination.

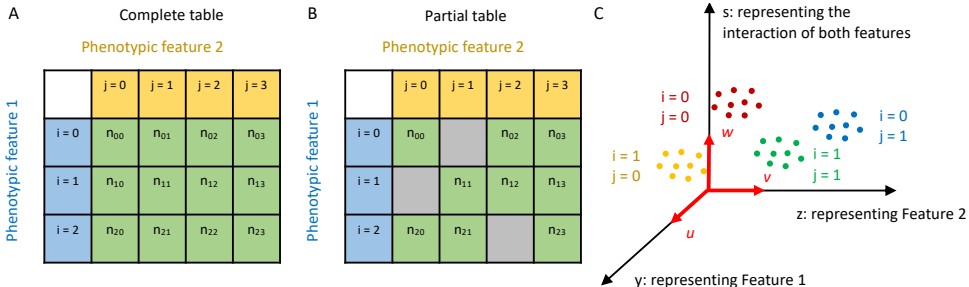

Figure 1: Illustration of our approach. (A,B) In the example, cell types are jointly represented by two phenotypic features, indexed with labels $i$ and $j$ respectively. If only some combinations of the two features are observed, one obtains a partial contingency table (B) instead of a complete one (A). (C) We seek linear projections of the data that separate the cell types in a factorized manner corresponding to the two features. Here $\boldsymbol{u}$, $\boldsymbol{v}$, and $\boldsymbol{w}$ are aligned with Feature 1, Feature 2, and the interaction of both features, with the projected coordinates $y$, $z$, and $s$ respectively.

## 2 FACTORIZED LINEAR DISCRIMINANT ANALYSIS (FLDA)

Suppose that we are given gene expression data of single neurons which are typically very high-dimensional. These cells are classified into cell types, as a result of clustering in the high-dimensional space and annotations based on prior knowledge or verification outcome (Macosko et al., 2015; Tasic et al., 2016; Shekhar et al., 2016; Tasic et al., 2018; Peng et al., 2019). We know the phenotypic traits of each neuronal type, therefore each type can also be jointed defined by the phenotypic features. We want to find an interpretable low-dimensional embedding in which certain dimensions represent factors of phenotypic features or their interactions. This requires that variation along one of the axes in the embedding space causes the variation of only one factor. In reality, this is hard to satisfy due to noise in the data, and we relax the constraint by letting data projected along one axis vary largely with one factor while minimally with the others. In addition, we ask that cells classified as the same type are still close to each other in the embedding space, while cells of different types are far apart.

As a start, let us consider only two phenotypic features of neurons, dendritic stratification, and axonal termination, both of which can be described with discrete categories, such as different regions or layers in the brain (Oh et al., 2014; Euler et al., 2014; Sanes & Masland, 2015; Kurmangaliyev et al., 2019). Suppose that each cell type can be jointly represented by its dendritic location indexed as $i$ and axonal lamination indexed as $j$, with the number of cells within each cell type $n_{ij}$. This representation can be described using a contingency table (Figure 1A,B). Note here that we allow the table to be partially filled.

Let $\boldsymbol{x}_{ijk}(k \in 1, 2, ...n_{ij})$ represent the expression values of $g$ genes in each cell ($\boldsymbol{x}_{ijk} \in \mathbf{R}^g$). How to find linear projections $y_{ijk} = \boldsymbol{u}^T \boldsymbol{x}_{ijk}$ and $z_{ijk} = \boldsymbol{v}^T \boldsymbol{x}_{ijk}$ that are aligned with features $i$ and $j$ respectively (Figure 1C)? We first asked whether we could factorize, for example, $y_{ijk}$, with respect to components depending on features $i$ and $j$. Indeed, motivated by the linear factor models used in multi-way ANOVA and the idea of partitioning variance, we constructed an objective function as the following, and found $\boldsymbol{u}^*$ that maximizes the objective (see detailed analysis in Appendix A):

$$\boldsymbol{u}^* = \underset{\boldsymbol{u} \in \mathbf{R}^g}{\arg \max} \frac{\boldsymbol{u}^T \boldsymbol{N}_A \boldsymbol{u}}{\boldsymbol{u}^T \boldsymbol{M}_e \boldsymbol{u}} \tag{1}$$

When we have a complete table, and there are $a$ levels for the feature $i$ and $b$ levels for the feature $j$, we have

$$\boldsymbol{N}_A = \boldsymbol{M}_A - \lambda_1 \boldsymbol{M}_B - \lambda_2 \boldsymbol{M}_{AB} \tag{2}$$

where $\boldsymbol{M}_A$, $\boldsymbol{M}_B$, and $\boldsymbol{M}_{AB}$ are the covariance matrices explained by the feature $i$, the feature $j$, and the interaction of them. $\lambda_1$ and $\lambda_2$ are hyper-parameters controlling the relative weights of $\boldsymbol{M}_B$

and $M_{AB}$ with respect to $M_A$. $M_e$ is the residual covariance matrix representing noise in gene expressions. Formal definitions of these terms are the following:

$$M_A = \frac{1}{a-1} \sum_{i=1}^{a} \sum_{j=1}^{b} (m_{i\cdot} - m_{\cdot\cdot})(m_{i\cdot} - m_{\cdot\cdot})^T \tag{3}$$

$$M_B = \frac{1}{b-1} \sum_{i=1}^{a} \sum_{j=1}^{b} (m_{\cdot j} - m_{\cdot\cdot})(m_{\cdot j} - m_{\cdot\cdot})^T \tag{4}$$

$$M_{AB} = \frac{1}{(a-1)(b-1)} \sum_{i=1}^{a} \sum_{j=1}^{b} (m_{ij} - m_{i\cdot} - m_{\cdot j} + m_{\cdot\cdot})(m_{ij} - m_{i\cdot} - m_{\cdot j} + m_{\cdot\cdot})^T \tag{5}$$

$$M_e = \frac{1}{N-ab} \sum_{i=1}^{a} \sum_{j=1}^{b} [\frac{1}{n_{ij}} \sum_{k=1}^{n_{ij}} (x_{ijk} - m_{ij})(x_{ijk} - m_{ij})^T] \tag{6}$$

where

$$m_{\cdot\cdot} = \frac{1}{ab} \sum_{i=1}^{a} \sum_{j=1}^{b} m_{ij} \tag{7}$$

$$m_{i\cdot} = \frac{1}{b} \sum_{j=1}^{b} m_{ij} \tag{8}$$

$$m_{\cdot j} = \frac{1}{a} \sum_{i=1}^{a} m_{ij} \tag{9}$$

in which

$$m_{ij} = \frac{1}{n_{ij}} \sum_{k=1}^{n_{ij}} x_{ijk} \tag{10}$$

An analogous expression provides the linear projection $v^*$ for the feature $j$, and $w^*$ for the interaction of both features $i$ and $j$. Similar arguments can be applied to the scenario of a partial table to find $u^*$ or $v^*$ as the linear projection for the feature $i$ or $j$ (see Appendix B for mathematical details).

Note that $N_A$ is symmetric and $M_e$ is positive definite. Therefore the optimization problem is a generalized eigenvalue problem (Ghojogh et al., 2019). When $M_e$ is invertible, $u^*$ is the eigenvector associated with the largest eigenvalue of $M_e^{-1} N_A$. In general, if we want to embed $x_{ijk}$ into a $d$-dimensional subspace aligned with the feature $i$ ($d < a$), we can take the eigenvectors with the $d$ largest eigenvalues of $M_e^{-1} N_A$, which we call the top $d$ factorized linear discriminant components (FLDs). Since multi-way ANOVA can handle contingency tables with more than two dimensions, our analysis can be easily generalized to more than two features.

## 3 Sparsity-based regularization of FLDA

For this domain-specific application in neurobiology, there is particular interest in finding a small group of genes that best determine one of the phenotypic features. This leads to finding axes that have only a few non-zero components. To identify such a sparse solution, we solved the following optimization problem:

$$\boldsymbol{u}^* = \arg\max_{\boldsymbol{u} \in \mathbf{R}^g} \frac{\boldsymbol{u}^T \boldsymbol{N}_A \boldsymbol{u}}{\boldsymbol{u}^T \boldsymbol{M}_e \boldsymbol{u}} \quad \text{subject to} \quad ||\boldsymbol{u}||_0 \leq l \tag{11}$$

from which the number of non-zero elements of $\boldsymbol{u}^*$ is less or equal to $l$.

This is known as a sparse generalized eigenvalue problem, which has three unique challenges, as listed in Tan et al. (2018): first, when the data are very high-dimensional, $\boldsymbol{M}_e$ can be singular and non-invertible; second, because of the normalization term $\boldsymbol{u}^T \boldsymbol{M}_e \boldsymbol{u}$, many solutions for sparse eigenvalue problems cannot be applied directly; finally, this problem involves maximizing a convex objective over a nonconvex set, which is NP-hard.

To solve it, we used truncated Rayleigh flow (Rifle), a method specifically developed to solve sparse generalized eigenvalue problems. The algorithm of Rifle is composed of two steps (Tan et al., 2018): first, to obtain an initial vector $\boldsymbol{u}_0$ that is close to $\boldsymbol{u}^*$. We used the solution from the non-sparse FLDA as an initial estimate of $\boldsymbol{u}_0$; second, iteratively, to perform a gradient ascent step on the objective function, and then execute a truncation step that preserves the $l$ entries of $\boldsymbol{u}$ with the largest values and sets the remaining entries to 0. Pseudo-code for this algorithm can be found in Appendix C.

As proved in Tan et al. (2018), if there is a unique sparse leading generalized eigenvector, Rifle will converge linearly to it with the optimal statistical rate of convergence. The computational complexity of the second step is $O(lg + g)$ for each iteration, therefore Rifle scales linearly with $g$, the dimensionality of the original data. Based on the theoretical proof, to guarantee convergence, the hyperparameter $\eta$ was selected to be sufficiently small such that $\eta \lambda_{max}(\boldsymbol{M}_e) < 1$, where $\lambda_{max}(\boldsymbol{M}_e)$ is the largest eigenvalue of $\boldsymbol{M}_e$. In our case, the other hyperparameter $l$, indicating how many genes to be preserved, was empirically selected based on the design of a follow-up experiment. As mentioned later in Results, we chose $l$ to be 20, a reasonable number of candidate genes to be tested in a biological study.

## 4 RELATED WORK

### 4.1 LINEAR DIMENSIONALITY REDUCTION

FLDA is a method for linear dimensionality reduction (Cunningham & Ghahramani, 2015). Formally, linear dimensionality reduction is defined as the following: given $n$ data points each of $g$ dimensions, $\boldsymbol{X} = [\boldsymbol{x}_1, \boldsymbol{x}_2, ..., \boldsymbol{x}_n] \in \mathbf{R}^{g \times n}$, and a choice of reduced dimensionality $r < g$, optimize an objective function $f(.)$ to produce a linear projection $\boldsymbol{P} \in \mathbf{R}^{r \times g}$, and $\boldsymbol{Y} = \boldsymbol{P}\boldsymbol{X} \in \mathbf{R}^{r \times n}$ is the low-dimensional transformed data.

State-of-the-art methods for linear dimensionality reduction include principal component analysis (PCA), factor analysis (FA), linear multidimensional scaling (MDS), linear discriminant analysis (LDA), canonical correlations analysis (CCA), maximum autocorrelation factors (MAF), slow feature analysis (SFA), sufficient dimensionality reduction (SDR), locality preserving projections (LPP), and independent component analysis (ICA) (Cunningham & Ghahramani, 2015). Some of them are obviously unsuitable to the problem we want to solve, for example, MAF and SFA are developed for data with temporal structures (Larsen, 2002; Wiskott & Sejnowski, 2002), and LPP focuses on the local structures of the data instead of their global organization (He & Niyogi). The remaining approaches can be roughly grouped for either unsupervised or supervised linear dimensionality reduction, and we discuss them separately.

### 4.2 UNSUPERVISED METHODS FOR LINEAR DIMENSIONALITY REDUCTION

Unsupervised linear dimensionality reduction, including PCA (Jolliffe, 2002), ICA (Hyvärinen et al., 2001), FA (Spearman, 1904) and more, project data into a low-dimensional space without using supervision labels. They are not suitable to solve our problem, due to the specific characteristics of the gene expression data. The dimensionality of gene expression data is usually very high, with tens of thousands of genes, and expressions of a fair number of them can be very noisy. These noisy genes cause large variance among individual cells, but in an unstructured way. Without supervision signals from the phenotypic features, unsupervised methods tend to select these genes

to construct the low-dimensional space, which offers neither the desired alignment nor a good separation of cell type clusters. To illustrate this, we performed PCA on the gene expression data and compared it with FLDA. Briefly, we solved the following objective to find the linear projection:

$$\boldsymbol{u}^* = \arg\max_{\boldsymbol{u} \in \mathbf{R}^g} \frac{\boldsymbol{u}^T \boldsymbol{X} \boldsymbol{X}^T \boldsymbol{u}}{\boldsymbol{u}^T \boldsymbol{u}} \tag{12}$$

The outcome of this comparison is shown in Results.

### 4.3 Supervised Methods for Linear Dimensionality Reduction

Supervised linear dimensionality reduction, represented by LDA (Fisher, 1936; McLachlan, 2004) and CCA (Hotelling, 1936; Wang et al., 2016), can overcome the above issue. By including supervised signals of phenotypic features, we can devalue genes whose expressions are non-informative about the phenotypes.

#### 4.3.1 Linear Discriminant Analysis (LDA)

We name our method FLDA because its objective function has a similar format as that of LDA. LDA also models the difference among data organized in pre-determined classes. Formally, LDA solves the following optimization problem:

$$\boldsymbol{u}^* = \arg\max_{\boldsymbol{u} \in \mathbf{R}^g} \frac{\boldsymbol{u}^T \boldsymbol{\Sigma}_b \boldsymbol{u}}{\boldsymbol{u}^T \boldsymbol{\Sigma}_e \boldsymbol{u}} \tag{13}$$

where $\boldsymbol{\Sigma}_b$ and $\boldsymbol{\Sigma}_e$ are estimates of the between-class and within-class covariance matrices respectively.

Different from FLDA, the representation of these classes is not explicitly formulated as a contingency table composed of multiple features. The consequence is that, when applied to the example problem in which neuronal types are organized into a two-dimensional contingency table with phenotypic features $i$ and $j$, in general, axes from LDA are not aligned with these two phenotypic features.

However, in the example above, we can perform two separate LDAs for the two features. This allows the axes from each LDA to align with its specific feature. We call this approach "2LDAs". There are two limitations of this approach: first, it discards information about the component depending on the interaction of the two features which cannot be explained by a linear combination of them; second, it explicitly maximizes the segregation of cells with different feature levels which sometimes is not consistent with a good separation of cell type clusters. Detailed comparisons between LDA, "2LDAs" and FLDA can be found in Results.

#### 4.3.2 Canonical Correlation Analysis (CCA)

CCA projects two datasets $\boldsymbol{X}_a \in \mathbf{R}^{g \times n}$ and $\boldsymbol{X}_b \in \mathbf{R}^{d \times n}$ to $\boldsymbol{Y}_a \in \mathbf{R}^{r \times n}$ and $\boldsymbol{Y}_b \in \mathbf{R}^{r \times n}$, such that the correlation between $\boldsymbol{Y}_a$ and $\boldsymbol{Y}_b$ is maximized. Formally, it tries to maximize this objective:

$$(\boldsymbol{u}^*, \boldsymbol{v}^*) = \arg\max_{\boldsymbol{u} \in \mathbf{R}^g, \boldsymbol{v} \in \mathbf{R}^d} \frac{\boldsymbol{u}^T (\boldsymbol{X}_a \boldsymbol{X}_a^T)^{-\frac{1}{2}} \boldsymbol{X}_a \boldsymbol{X}_b^T (\boldsymbol{X}_b \boldsymbol{X}_b^T)^{-\frac{1}{2}} \boldsymbol{v}}{(\boldsymbol{u}^T \boldsymbol{u})^{-\frac{1}{2}} (\boldsymbol{v}^T \boldsymbol{v})^{-\frac{1}{2}}} \tag{14}$$

To apply CCA to our problem, we need to set $\boldsymbol{X}_a$ to be the gene expression matrix, and $\boldsymbol{X}_b$ to be the matrix of $d$ phenotypic features ($d = 2$ for two features as illustrated later). In contrast with FLDA, CCA finds a transformation of gene expressions aligned with a linear combination of phenotypic features, instead of a factorization of gene expressions corresponding to individual phenotypic features. This difference is quantified and shown in Results.

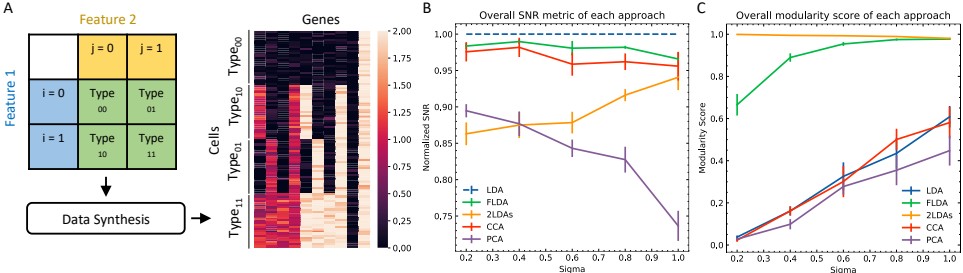

Figure 2: Quantitative comparison between FLDA and other models. (A) Illustration of data synthesis. See Appendix D for implementation details. Color bar indicates the expression values of the ten generated genes. (B) Normalized overall SNR metric of each analysis. The SNR values are normalized with respect to that of LDA. (C) Overall modularity score for each analysis. Error bars in (B,C) denote standard errors each calculated from 10 repeated simulations.

## 5    EXPERIMENTS

### 5.1    DATASETS

In order to quantitatively compare FLDA with other linear dimensionality reduction methods, such as PCA, CCA, LDA, and the "2LDAs" approach, we created synthetic datasets. Four types of cells, each containing 25 examples, were generated from a Cartesian product of two features $i$ and $j$, organized in a 2x2 complete contingency table. Expressions of 10 genes were generated for these cells, in which the levels of Genes 1-8 were correlated with either the feature $i$, the feature $j$, or the interactions of them, and the levels of the remaining 2 genes were purely driven by noise (Figure 2A). Details of generating the data can be found in Appendix D.

To illustrate FLDA in analyzing single-cell RNA-Seq datasets for real problems of neurobiology, and demonstrate the merit of our approach in selecting a few important genes for each phenotype, we used a dataset of Drosophila T4/T5 neurons (Kurmangaliyev et al., 2019). T4 and T5 neurons are very similar in terms of general morphology and physiological properties, but they differ by the location of their dendrites in the medulla and lobula, two distinct brain regions. T4 and T5 neurons each contain four subtypes, with each pair of the four laminating their axons in a specific layer in the lobula plate (Figure 3A). Therefore, we can use two phenotypic features to describe these neurons: the feature $i$ indicates the dendritic location at the medulla or lobula; the feature $j$ describes the axonal lamination at one of the four layers (a/b/c/d) (Figure 3B). In this experiment, we focused on the dataset containing expression data of 17492 genes from 3833 cells collected at a defined time during brain development.

### 5.2    DATA PREPROCESSING

The T4/T5 neuron dataset was preprocessed as previously reported (Shekhar et al., 2016; Peng et al., 2019; Tran et al., 2019). Briefly, transcript counts within each column of the count matrix (genes×cells) were normalized to sum to the median number of transcripts per cell, resulting in the normalized counts Transcripts-per-median or $TPM_{gc}$ for Gene $g$ in Cell $c$. We used the log-transformed expression data $E_{gc} = \ln(TPM_{gc} + 1)$ for further analysis. We adopted a common approach in single-cell RNA-Seq studies that is based on fitting a relationship between mean and coefficient of variation (Chen et al., 2016; Pandey et al., 2018) to select highly variable genes, and performed FLDA on the expression data with only these genes. We preprocessed the data with PCA and kept principal components (PCs) explaining ∼99% of the total variance before running FLDA but not the sparse version of the algorithm. In the experiment below, we set the hyper-parameters $\lambda$s in Equation (2) to 1.

### 5.3    METRICS

We included the following metrics to evaluate our method: A signal-to-noise ratio (SNR) measures how well each discriminant axis separates cell types compared with noise estimated from the vari-

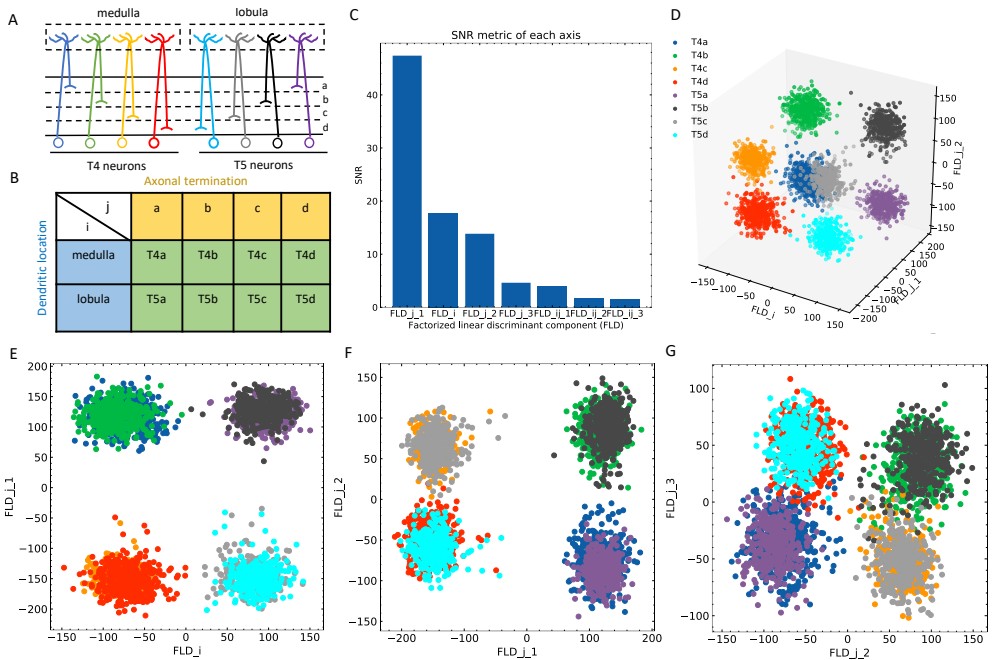

Figure 3: FLDA on the dataset of T4/T5 neurons. (A) T4/T5 neuronal types and their dendritic and axonal phenotypes. (B) T4/T5 neurons can be organized in a complete contingency table. Here $i$ indicates the dendritic location and $j$ indicates the axonal termination. (C) SNR metric of each discriminant axis. (D) Projection of the data into the three-dimensional space consisting of the discriminant axis for the feature $i$ (FLD$_i$) and the first and second discriminant axes for the feature $j$ (FLD$_{j1}$ and FLD$_{j2}$). (E-G) Projection of the data into the two-dimensional space made of FLD$_i$ and FLD$_{j1}$ (E), FLD$_{j1}$ and FLD$_{j2}$ (F), or FLD$_{j2}$ and FLD$_{j3}$ (the third discriminant axis for the feature $j$) (G). Different cell types are indicated by different colors as in (A) and (D).

ance within cell type clusters. The explained variance (EV) for each discriminant axis measures how much variance of the feature $i$ or $j$ is explained among the total variance explained by that axis. The mutual information (MI) between each discriminant axis and each feature quantifies how "informative" an axis is to a specific feature. Built on the calculation of MI, we included the modularity score which measures whether each discriminant axis depends on at most one feature (Ridgeway & Mozer, 2018). The implementation details of these metrics can be found in Appendix E.

## 6   RESULTS

To quantitatively compare the difference between FLDA and other alternative models including PCA, CCA, LDA, and "2LDAs", we measured the proposed metrics from analyses of the synthesized datasets (Figure 2A). Given that the synthesized data were organized in a 2x2 contingency table, each LDA of the "2LDAs" approach could find only one dimension for the specific feature $i$ or $j$. Therefore, as a fair comparison, we only included the corresponding dimensions in FLDA (FLD$_i$ and FLD$_j$) and the top two components of PCA, CCA, and LDA. The overall SNR values normalized by that of LDA and the overall modularity scores were plotted for data generated with different noise levels (Figure 2B,C). The performance of PCA is the worst among all these models because the unsupervised approach cannot prevent the noise from contaminating the signal. The supervised approaches in general have good SNRs, but LDA and CCA suffer from low modularity scores. This is expected because LDA maximizes the separation of cell type clusters but overlooks the alignment of the axes to the feature $i$ or $j$, and CCA maximizes the correlation to a linear combination of phenotypic features instead of individual ones. By contrast, "2LDAs" achieves the highest modularity scores but has the worst SNR among the supervised approaches, because it tries to maximize the separation of cells with different feature levels, which is not necessarily consistent with maximizing the segregation of cell types. Both the SNR value and the modularity score of FLDA

are close to the optimal, as it not only considers the alignment of axes to different features but also constrains the variance within cell types. A representative plot of the EV and MI metrics of these models is shown in Figure 5, reporting good alignment of axes to either the feature $i$ or $j$ in FLDA and '2LDAs'', but not in the others.

A question of significance in neurobiology is whether the diverse phenotypes of neuronal cell types are generated by combinations of modular transcriptional programs, and if so, what is the gene signature for each of the programs. To illustrate the ability of our approach in addressing this problem, we applied FLDA to the dataset of Drosophila T4/T5 neurons. The T4/T5 neurons could be organized in a 2x4 contingency table, therefore, FLDA was able to project the expression data into a subspace of seven dimensions, with one FLD aligned with dendritic location $i$ ($FLD_i$), three FLDs aligned with axonal termination $j$ ($FLD_{j1-3}$), and the remaining three representing the interaction of both phenotypes ($FLD_{ij1-3}$). We ranked these axes based on their SNR metrics and found that $FLD_{j1}$, $FLD_i$, and $FLD_{j2}$ have much higher SNRs than the rest (Figure 3C). Indeed, data representations in the subspace consisting of these three dimensions show a clear separation of the eight neuronal types (Figure 3D). As expected, $FLD_i$ teases apart T4 from T5 neurons, whose dendrites are located at different brain regions (Figure 3E). Interestingly, $FLD_{j1}$ separates T4/T5 neurons into two groups, a/b vs c/d, corresponding to the upper or lower lobula place, and $FLD_{j2}$ divides them into another two, a/d vs b/c, indicating whether their axons laminate at the middle or lateral part of the lobula plate (Figure 3E,F). Unexpectedly, among these three dimensions, $FLD_{j1}$ has a much higher SNR than $FLD_i$ and $FLD_{j2}$, whose SNR values are similar. This suggests a hierarchical structure in the genetic organization of T4/T5 neurons: they are first separated into either a/b or c/d types, and subsequently divided into each of the eight subtypes. In fact, this exactly matches the sequence of their cell fate determination during development, as revealed in a previous genetic study (Pinto-Teixeira et al., 2018). Finally, the last discriminant axis of the axonal feature $FLD_{j3}$ separates the group a/c from b/d, suggesting its role in fine-tuning the axonal depth within the upper or lower lobula plate (Figure 3G).

To seek gene signatures for the discriminant components in FLDA, we applied the sparsity-based regularization to constrain the number of genes with non-zero weight coefficients. Here we set the number to 20, a reasonable number of candidate genes that might be tested in a follow-up biological study. We extracted a list of 20 genes each for the axis of $FLD_i$ or $FLD_{j1}$. The relative importance of these genes to each axis is directly informed by their weight values (Figure 4A,C). Side-by-side, we plotted expression profiles of these genes in the eight neuronal types (Figure 4B,D). For both axes, the genes critical in separating cells with different feature levels are differentially expressed in corresponding cell types. We compared our gene lists with those obtained using conventional methods which were reported in Kurmangaliyev et al. (2019). Consistent with the report, we found indicator genes for dendritic location such as $TfAP$-$2$, $dpr2$, $CG34155$, and $CG12065$, and those for axonal lamination such as $klg$, $bi$, $pros$. In addition, we found genes that were not reported in this previous study. For example, our results suggest that the genes $Thor$ and $pHCl$-$1$ are important to the dendritic phenotype, and $Lac$ and $Mip$ are critical to the axonal phenotype. These are promising genetic targets to be tested in biological experiments. Lastly, FLDA allowed us to examine the component that depends on the interaction of both features and identify its gene signature, which provides clues to transcriptional regulation of gene expressions in the T4/T5 neuronal types (Figures 6 and 7).

As a supervised approach, FLDA depends on correct phenotype labels to extract meaningful information. But if the phenotypes are annotated incorrectly, can we use FLDA to raise a flag? We propose a perturbation analysis of FLDA to address this question built on the assumption that among possible phenotype annotations, the projection of gene expression data based on correct labels leads to better metric measurements than incorrect ones. As detailed in Appendix F, we generated three kinds of incorrect labels for the dataset of T4/T5 neurons, corresponding to three common scenarios of mislabeling: the phenotypes of a cell type were mislabeled with those of another type; a singular phenotypic level was incorrectly split into two; two phenotypic levels are incorrectly merged into one. FLDA was applied to gene expressions of T4/T5 neurons but with these perturbed annotations. Proposed metrics such as the SNR value and modularity score were plotted in Figure 8. Indeed, the projection of gene expressions with correct annotation leads to the best SNR value and modularity score compared with incorrect annotations. This implies that this type of perturbation analysis is a useful practice in general: it raises the confidence that the original annotation is correct if FLDA on the perturbed annotations produces lower metric scores.

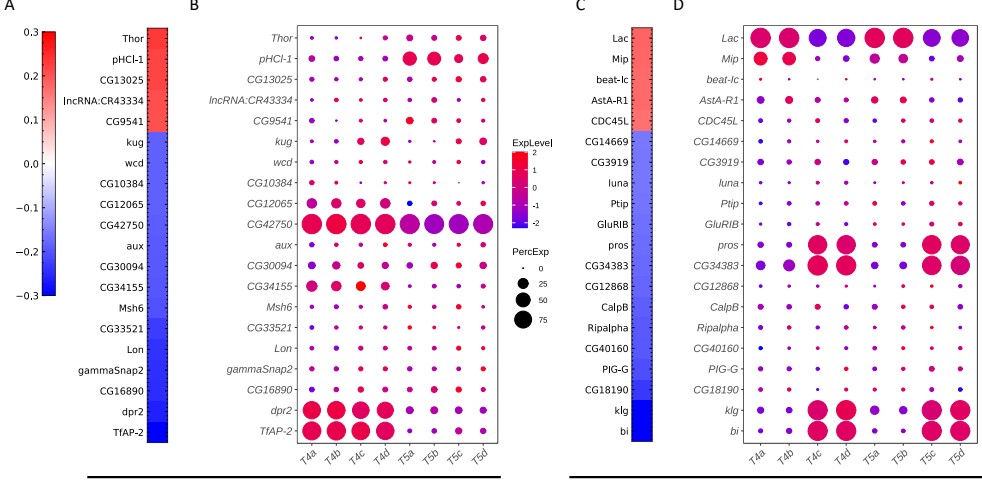

Figure 4: Critical genes extracted from the sparse algorithm. (A) Weight vector of the 20 genes selected for the dendritic phenotype (FLD$_i$). The weight value is indicated in the color bar with color indicating direction (red: positive and green: negative) and saturation indicating magnitude. (B) Expression patterns of the 20 genes from (A) in eight types of T4/T5 neurons. Dot size indicates the percentage of cells in which the gene was expressed, and color represents average scaled expression. (C) Weight vector of the 20 genes selected for the axonal phenotype (FLD$_{j1}$). Legend as in (A). (D) Expression patterns of the 20 genes from (C) in eight types of T4/T5 neurons. Legend as in (B).

# 7    DISCUSSION

We developed FLDA, a novel supervised linear dimensionality reduction method for understanding the relationship between high-dimensional gene expression patterns and cellular phenotypes. We illustrate the power of FLDA by analyzing a gene expression dataset from Drosophila T4/T5 neurons that are labeled by two phenotypic features, each with multiple levels. The approach allowed us to identify new genes for each of the phenotypic features that were not apparent under conventional methods. Furthermore, we found a hierarchical relationship in the genetic organization of these cells. These findings point the way for new biological experiments.

The approach is motivated by multi-way ANOVA, and thus it generalizes easily to more than two features. Future applications in neurobiology include the analysis of phenotypic characteristics such as electrophysiology and connectivity (Zeng & Sanes, 2017; Gouwens et al., 2019; 2020). More generally FLDA can be applied to any labeled data set for which the labels form a Cartesian product of multiple features. For example, this would include face images that can be jointly labeled by the age, gender, and other features of a person (Moghaddam & Ming-Hsuan Yang, 2002; Zhang et al., 2017).

FLDA factorizes gene expression data into features and their interactions, and finds a linear projection of the data that varies with only one factor but not the others. This provides a modular representation aligned with the factors (Bengio et al., 2012). Ridgeway & Mozer (2018) argued that modularity together with explicitness could define disentangled representations. Our approach is linear, which presents an explicit mapping between gene expressions and phenotypic features, therefore our approach can potentially serve as a supervised approach to disentanglement (Kingma et al., 2014; Kulkarni et al., 2015; Karaletsos et al., 2016).

Compared with other non-linear embedding methods for cell types (Hinton & Salakhutdinov, 2006; Feng et al., 2014; Gala et al., 2019), the linear nature of FLDA makes it extremely easy to interpret the low-dimensional representations, as the weight vector directly informs the relative importance of each gene. To allow the selection of a small set of critical genes, we leveraged our approach with sparse regularization. This makes FLDA especially useful to experimentalists who can take the list of genes and test them in a subsequent round of experiments.

# 8 APPENDIX

## 8.1 A. OBJECTIVE FUNCTIONS

Here we derive the objective functions used in our analysis. Again if $\boldsymbol{x}_{ijk}(k \in 1, 2, ...n_{ij})$ represents the expression values of $g$ genes in each cell ($\boldsymbol{x}_{ijk} \in \mathbf{R}^g$)), we seek to find a linear projection $y_{ijk} = \boldsymbol{u}^T \boldsymbol{x}_{ijk}$ that is aligned with the feature $i$.

### 8.1.1 INSPIRATION FROM ANOVA

We asked what is the best way to factorize $y_{ijk}$. Inspired by multi-way ANOVA (Fisher, 1918), we identified three components: one depending on the feature $i$, another depending on the feature $j$, and the last one depending on the interaction of both features. We therefore followed the procedures of ANOVA to partition sums of squares and factorize $y_{ijk}$ into these three components.

Let us first assume that all cell types defined by $i$ and $j$ contain the same number of cells. With cell types represented by a complete contingency table (Figure 1A), $y_{ijk}$ can be linearly factorized using the model of two crossed factors. Formally, the linear factorization is the following:

$$y_{ijk} = \mu + \alpha_i + \beta_j + (\alpha\beta)_{ij} + \epsilon_{ijk} \tag{15}$$

where $y_{ijk}$ represents the coordinate of the $k$th cell in the category defined by $i$ and $j$; $\mu$ is the average level of $y$; $\alpha_i$ is the component that depends on the feature $i$, and $\beta_j$ is the component that depends on the feature $j$; $(\alpha\beta)_{ij}$ describes the component that depends on the interaction of both features $i$ and $j$; $\epsilon_{ijk} \sim \mathcal{N}(0, \sigma^2)$ is the residual of this factorization.

Let us say that the features $i$ and $j$ fall into $a$ and $b$ discrete categories respectively. Then without loss of generality, we can require:

$$\sum_{i=1}^{a} \alpha_i = 0 \tag{16}$$

$$\sum_{j=1}^{b} \beta_j = 0 \tag{17}$$

$$\sum_{i=1}^{a} (\alpha\beta)_{ij} = \sum_{j=1}^{b} (\alpha\beta)_{ij} = 0 \tag{18}$$

Corresponding to these, there are three null hypotheses:

$$H_{01} : \alpha_i = 0 \tag{19}$$

$$H_{02} : \beta_j = 0 \tag{20}$$

$$H_{03} : (\alpha\beta)_{ij} = 0 \tag{21}$$

Here we want to reject $H_{01}$ while accepting $H_{02}$ and $H_{03}$ in order that $y_{ijk}$ is aligned with the feature $i$.

Next, we partition the total sum of squares. If the number of cells within each cell type category is $n$, and the total number of cells is $N$, then we have

$$\sum_{i=1}^{a}\sum_{j=1}^{b}\sum_{k=1}^{n}(y_{ijk} - \bar{y}_{...})^2 = bn\sum_{i=1}^{a}(\bar{y}_{i..} - \bar{y}_{...})^2 + an\sum_{j=1}^{b}(\bar{y}_{.j.} - \bar{y}_{...})^2$$
$$+ n\sum_{i=1}^{a}\sum_{j=1}^{b}(\bar{y}_{ij.} - \bar{y}_{i..} - \bar{y}_{.j.} + \bar{y}_{...})^2 + \sum_{i=1}^{a}\sum_{j=1}^{b}\sum_{k=1}^{n}(y_{ijk} - \bar{y}_{ij.})^2 \tag{22}$$

where $\bar{y}$ is the average of $y_{ijk}$ over the indices indicated by the dots. Equation (22) can be written as

$$SS_T = SS_A + SS_B + SS_{AB} + SS_e \tag{23}$$

with each term having degrees of freedom $N-1$, $a-1$, $b-1$, $(a-1)(b-1)$, and $N-ab$ respectively. Here $SS_A$, $SS_B$, $SS_{AB}$, and $SS_e$ are partitioned sum of squares for the factors $\alpha_i$, $\beta_j$, $(\alpha\beta)_{ij}$, and the residual.

ANOVA rejects or accepts a null hypothesis by comparing its mean square (the partitioned sum of squares normalized by the degree of freedom) to that of the residual. This is done by constructing F-statistics for each factor as shown below:

$$F_A = \frac{MS_A}{MS_e} = \frac{\frac{SS_A}{a-1}}{\frac{SS_e}{N-ab}} \tag{24}$$

$$F_B = \frac{MS_B}{MS_e} = \frac{\frac{SS_B}{b-1}}{\frac{SS_e}{N-ab}} \tag{25}$$

$$F_{AB} = \frac{MS_{AB}}{MS_e} = \frac{\frac{SS_{AB}}{(a-1)(b-1)}}{\frac{SS_e}{N-ab}} \tag{26}$$

Under the null hypotheses, the F-statistics follow the F-distribution. Therefore, a null hypothesis is rejected when we observe the value of a F-statistic above a certain threshold calculated from the F-distribution. Here we want $F_A$ to be large enough so that we can reject $H_{01}$, but $F_B$ and $F_{AB}$ to be small enough for us to accept $H_{02}$ and $H_{03}$. In other words, we want to maximize $F_A$ while minimizing $F_B$ and $F_{AB}$. Therefore, we propose maximizing an objective $L$:

$$L = F_A - \lambda_1 F_B - \lambda_2 F_{AB} \tag{27}$$

where $\lambda_1$ and $\lambda_2$ are hyper-parameters determining the relative weights of $F_B$ and $F_{AB}$ compared with $F_A$.

### 8.1.2 Objective functions under a complete contingency table

When the numbers of cells within categories defined by $i$ and $j$ ($n_{ij}$) are not all the same, the total sum of squares cannot be partitioned as in Equation (22). However, if we only care about distinctions between cell types instead of individual cells, we can use the mean value of each cell type cluster ($\bar{y}_{ij.}$) to estimate the overall average value ($\tilde{y}_{...}$), and the average value of each category $i$ ($\tilde{y}_{i..}$) or $j$ ($\tilde{y}_{.j.}$). Therefore, Equation (22) can be modified as the following:

$$\sum_{i=1}^{a}\sum_{j=1}^{b}[\frac{1}{n_{ij}}\sum_{k=1}^{n_{ij}}(y_{ijk} - \tilde{y}_{...})^2] = b\sum_{i=1}^{a}(\tilde{y}_{i..} - \tilde{y}_{...})^2 + a\sum_{j=1}^{b}(\tilde{y}_{.j.} - \tilde{y}_{...})^2$$
$$+ \sum_{i=1}^{a}\sum_{j=1}^{b}(\bar{y}_{ij.} - \tilde{y}_{i..} - \tilde{y}_{.j.} + \tilde{y}_{...})^2 + \sum_{i=1}^{a}\sum_{j=1}^{b}[\frac{1}{n_{ij}}\sum_{k=1}^{n_{ij}}(y_{ijk} - \bar{y}_{ij.})^2] \tag{28}$$

where

$$\bar{y}_{ij.} = \frac{\sum_{k=1}^{n_{ij}} y_{ijk}}{n_{ij}} \tag{29}$$

$$\tilde{y}_{i..} = \frac{\sum_{j=1}^{b} \bar{y}_{ij.}}{b} \tag{30}$$

$$\tilde{y}_{.j.} = \frac{\sum_{i=1}^{a} \bar{y}_{ij.}}{a} \tag{31}$$

$$\tilde{y}_{...} = \frac{\sum_{i=1}^{a} \sum_{j=1}^{b} \bar{y}_{ij.}}{ab} \tag{32}$$

If we describe Equation (28) as:

$$\tilde{SS}_T = \tilde{SS}_A + \tilde{SS}_B + \tilde{SS}_{AB} + \tilde{SS}_e \tag{33}$$

then following the same arguments, we want to maximize an objective function in the following format:

$$L = \frac{\frac{\tilde{SS}_A}{a-1} - \lambda_1 \frac{\tilde{SS}_B}{b-1} - \lambda_2 \frac{\tilde{SS}_{AB}}{(a-1)(b-1)}}{\frac{\tilde{SS}_e}{N-ab}} \tag{34}$$

### 8.1.3 OBJECTIVE FUNCTIONS UNDER A PARTIAL CONTINGENCY TABLE

When we have a representation of a partial table, we can no longer separate out the component that depends on the interaction of both features. Therefore, we use another model, a linear model of two nested factors, to factorize $y_{ijk}$, which has the following format:

$$y_{ijk} = \mu + \alpha_i + \beta_{j(i)} + \epsilon_{ijk} \tag{35}$$

Note that we now have $\beta_{j(i)}$ instead of $\beta_j + (\alpha\beta)_{ij}$. In this model, we identify a primary factor, for instance, the feature denoted by $i$ which falls into $a$ categories, and the other (indexed by $j$) becomes a secondary factor, the number of whose levels $b_i$ depends on the level of the primary factor. We merge the component depending on the interaction of both features into that of the secondary factor as $\beta_{j(i)}$.

Similarly, we have

$$\sum_{i=1}^{a} \sum_{j=1}^{b_i} [\frac{1}{n_{ij}} \sum_{k=1}^{n_{ij}} (y_{ijk} - \tilde{y}_{...})^2] = \sum_{i=1}^{a} [\sum_{j=1}^{b_i} (\tilde{y}_{i..} - \tilde{y}_{...})^2]$$
$$+ \sum_{i=1}^{a} \sum_{j=1}^{b_i} (\bar{y}_{ij.} - \tilde{y}_{i..})^2 + \sum_{i=1}^{a} \sum_{j=1}^{b_i} [\frac{1}{n_{ij}} \sum_{k=1}^{n_{ij}} (y_{ijk} - \bar{y}_{ij.})^2] \tag{36}$$

which can be written as

$$\tilde{SS}_T = \tilde{SS}_A + \tilde{SS}_B + \tilde{SS}_e \tag{37}$$

with degrees of freedom $N-1$, $a-1$, $M-a$, and $N-M$ for each of the terms, where $M$ is:

$$M = \sum_{i=1}^{a} b_i \tag{38}$$

Therefore, we want to maximize the following objective:

$$L = \frac{\frac{\tilde{SS}_A}{a-1} - \lambda \frac{\tilde{SS}_B}{M-a}}{\frac{\tilde{SS}_e}{N-M}} \tag{39}$$

### 8.2 B. FLDA WITH A PARTIAL CONTINGENCY TABLE

Here we provide the mathematical details of FLDA under the representation of a partial table. When we have a partial table, if the feature $i$ is the primary feature with $a$ levels, and the feature $j$ is the secondary feature with $b_i$ levels, then $\boldsymbol{N}_A$ in Equation (1) is defined as follows:

$$\boldsymbol{N}_A = \boldsymbol{M}_A - \lambda \boldsymbol{M}_{B|A} \tag{40}$$

where

$$\boldsymbol{M}_A = \frac{1}{a-1} \sum_{i=1}^{a} \sum_{j=1}^{b_i} (\boldsymbol{m}_{i.} - \boldsymbol{m}_{..})(\boldsymbol{m}_{i.} - \boldsymbol{m}_{..})^T \tag{41}$$

$$\boldsymbol{M}_{B|A} = \frac{1}{M-a} \sum_{i=1}^{a} \sum_{j=1}^{b_i} (\boldsymbol{m}_{ij} - \boldsymbol{m}_{i.})(\boldsymbol{m}_{ij} - \boldsymbol{m}_{i.})^T \tag{42}$$

and $M$ is defined as in Equation (38). Correspondingly, $\boldsymbol{M}_e$ in Equation (1) is defined as:

$$\boldsymbol{M}_e = \frac{1}{N-M} \sum_{i=1}^{a} \sum_{j=1}^{b} [\frac{1}{n_{ij}} \sum_{k=1}^{n_{ij}} (\boldsymbol{x}_{ijk} - \boldsymbol{m}_{ij})(\boldsymbol{x}_{ijk} - \boldsymbol{m}_{ij})^T] \tag{43}$$

and

$$\boldsymbol{m}_{..} = \frac{1}{M} \sum_{i=1}^{a} \sum_{j=1}^{b_i} \boldsymbol{m}_{ij} \tag{44}$$

$$\boldsymbol{m}_{i.} = \frac{1}{b_i} \sum_{j=1}^{b_i} \boldsymbol{m}_{ij} \tag{45}$$

The remaining mathematical arguments are the same as those for the complete table. In this scenario, because we don't observe all possible combinations of features $i$ and $j$, we cannot find the linear projection for the interaction of both features.

### 8.3 C. PSEUDO-CODE FOR RIFLE

We show pseudo-code for the Rifle algorithm as follows:

---

**procedure** RIFLE($\boldsymbol{N}_A, \boldsymbol{M}_e, \boldsymbol{u}_0, l, \eta$) $\qquad\qquad\qquad\qquad\qquad$ ▷ $\eta$ is the step size
$\quad t = 1$ $\qquad\qquad\qquad\qquad\qquad\qquad\qquad\qquad\qquad$ ▷ $t$ indicates the iteration number
$\quad$ **while** not converge **do** $\qquad\qquad\qquad\qquad\qquad\qquad\qquad$ ▷ Converge when $\boldsymbol{u}_t \simeq \boldsymbol{u}_{t-1}$
$\qquad \rho_{t-1} \leftarrow \frac{\boldsymbol{u}_{t-1}^T \boldsymbol{N}_A \boldsymbol{u}_{t-1}}{\boldsymbol{u}_{t-1}^T \boldsymbol{M}_e \boldsymbol{u}_{t-1}}$
$\qquad \boldsymbol{C} \leftarrow \boldsymbol{I} + (\frac{\eta}{\rho_{t-1}})(\boldsymbol{N}_A - \rho_{t-1}\boldsymbol{M}_e)$
$\qquad \boldsymbol{u}_t \leftarrow \frac{\boldsymbol{C}\boldsymbol{u}_{t-1}}{||\boldsymbol{C}\boldsymbol{u}_{t-1}||_2}$
$\qquad$ Truncate $\boldsymbol{u}_t$ by keeping the top $l$ entries of $\boldsymbol{u}$ with the largest values and setting the rest
entries to 0
$\qquad \boldsymbol{u}_t \leftarrow \frac{\boldsymbol{u}_t}{||\boldsymbol{u}_t||_2}$
$\qquad t \leftarrow t + 1$
$\quad$ **end while**
$\quad$ **return** $\boldsymbol{u}_t$
**end procedure**

---

## 8.4 D. IMPLEMENTATION DETAILS OF DATA SYNTHESIS

To quantitatively compare FLDA with alternative approaches, we synthesized data of four cell types, each of which contained 25 cells. The four cell types were generated from a Cartesian product of two features $i$ and $j$, where $i \in \{0, 1\}$ and $j \in \{0, 1\}$. Expressions of 10 genes were generated for each cell. The expression value of the $h$th gene in the $k$th cell of the cell type $ij$, $g_{ijk}^h$ was defined as the following:

$$g_{ijk}^1 = i + \epsilon_{ijk} \tag{46}$$

$$g_{ijk}^2 = j + \epsilon_{ijk} \tag{47}$$

$$g_{ijk}^3 = i \wedge j + \epsilon_{ijk} \tag{48}$$

$$g_{ijk}^4 = i \vee j + \epsilon_{ijk} \tag{49}$$

$$g_{ijk}^5 = 2i + \epsilon_{ijk} \tag{50}$$

$$g_{ijk}^6 = 2j + \epsilon_{ijk} \tag{51}$$

$$g_{ijk}^7 = 2i \wedge j + \epsilon_{ijk} \tag{52}$$

$$g_{ijk}^8 = 2i \vee j + \epsilon_{ijk} \tag{53}$$

$$g_{ijk}^9 = \epsilon_{ijk} \tag{54}$$

$$g_{ijk}^{10} = 2 + \epsilon_{ijk} \tag{55}$$

where

$$i \wedge j = \begin{cases} 1, & \text{if } i = 1, j = 1 \\ 0, & \text{otherwise} \end{cases} \tag{56}$$

and

$$i \vee j = \begin{cases} 0, & \text{if } i = 0, j = 0 \\ 1, & \text{otherwise} \end{cases} \tag{57}$$

were interactions of the two features. Here $\epsilon_{ijk}$ was driven by Gaussian noise, namely,

$$\epsilon_{ijk} \sim \mathcal{N}(0, \sigma^2) \tag{58}$$

We synthesized datasets of 5 different $\sigma$ values ($\sigma \in \{0.2, 0.4, 0.6, 0.8, 1.0\}$). This was repeated 10 times and metrics for each $\sigma$ value were calculated as the average across the 10 repeats.

## 8.5    E. IMPLEMENTATION DETAILS OF THE METRICS USED IN THE STUDY

We measured the following metrics in our experiments:

### 8.5.1    SIGNAL-TO-NOISE RATIO (SNR)

Because we care about the separation of cell types, we define the SNR metric as the ratio of the variance between cell types over the variance of the noise, which is estimated from within-cluster variance. For the entire embedding space, given $q$ cell types, if the coordinate of each cell is indicated by $c$, then we define the overall SNR metric as the following:

$$SNR_{overall} = \frac{tr(\Sigma_{p=1}^{q} n_p (\bar{c}_{p.} - \bar{c}_{..})(\bar{c}_{p.} - \bar{c}_{..})^T))}{tr(\Sigma_{p=1}^{q} \Sigma_{k=1}^{n_p} (c_{pk} - \bar{c}_{p.})(c_{pk} - \bar{c}_{p.})^T)} \tag{59}$$

where $\bar{c}_{p.}$ is the center of each cell type cluster, and $\bar{c}_{..}$ is the center of all data points.

Let $c$ denote the embedded coordinate along a specific dimension. The SNR metric for that axis is therefore:

$$SNR = \frac{\Sigma_{p=1}^{q} n_p (\bar{c}_{p.} - \bar{c}_{..})^2}{\Sigma_{p=1}^{q} \Sigma_{k=1}^{n_p} (c_{pk} - \bar{c}_{p.})^2} \tag{60}$$

### 8.5.2    EXPLAINED VARIANCE (EV)

We want to know whether the variation of a specific dimension is strongly explained by that of a specific feature. Therefore, we measure, for each axis, how much of the total explained variance is explained by the variance of the feature $i$ or $j$. Formally, given the embedded coordinate $y_{ijk}$, we calculate the EV as the following:

$$EV_i = \frac{\sum_{i=1}^{a} \sum_{j=1}^{b} n_{ij} (\bar{y}_{i..} - \bar{y}_{...})^2}{\sum_{i=1}^{a} \sum_{j=1}^{b} \sum_{k=1}^{n_{ij}} (y_{ijk} - \bar{y}_{...})^2} \tag{61}$$

$$EV_j = \frac{\sum_{i=1}^{a} \sum_{j=1}^{b} n_{ij} (\bar{y}_{.j.} - \bar{y}_{...})^2}{\sum_{i=1}^{a} \sum_{j=1}^{b} \sum_{k=1}^{n_{ij}} (y_{ijk} - \bar{y}_{...})^2} \tag{62}$$

where $\bar{y}$ is the average of $y_{ijk}$ over the indices indicated by the dots.

### 8.5.3    MUTUAL INFORMATION (MI)

The MI between a discriminant axis $u$ and a feature quantifies how much information of the feature is obtained by observing data projected along that axis. It is calculated as the MI between data representations along the axis $y = u^T X$ and feature labels of the data $f$, where $X$ is the original gene expression matrix:

$$I(\boldsymbol{y}, \boldsymbol{f}) = H(\boldsymbol{y}) + H(\boldsymbol{f}) - H(\boldsymbol{y}, \boldsymbol{f})$$
$$= -\sum_{y \in Y} p(y) \log_2 p(y) - \sum_{f \in F} p(f) \log_2 p(f) - \sum_{y \in Y} \sum_{f \in F} p(y, f) \log_2 p(y, f) \qquad (63)$$

Here $H$ indicates entropy. To calculate $H(\boldsymbol{y})$ and $H(\boldsymbol{y}, \boldsymbol{f})$, we discretize $\boldsymbol{y}$ into 10 bins.

### 8.5.4 MODULARITY

Ridgeway & Mozer (2018) argued that in a modular representation, each axis should depend on at most a single feature. Following the arguments in their paper, the modularity score is computed as follows: we first calculate the MI between each feature and each axis ($m_{if}$ denotes the MI between one axis $i$ and one feature $f$). If an axis is perfectly modular, it will have high mutual information for only one feature and zeros for the others, we therefore compute a template $t_{if}$ as the following:

$$t_{if} = \begin{cases} \theta_i, & \text{if } f = \arg\max_g m_{ig} \\ 0, & \text{otherwise} \end{cases} \qquad (64)$$

where $\theta_i = \max_g m_{ig}$. We then calculate the deviation from the template as:

$$\delta_i = \frac{\sum_f (m_{if} - t_{if})^2}{\theta_i^2 (N - 1)} \qquad (65)$$

where $N$ is the number of features. The modularity score for the axis $i$ is $1 - \delta_i$. The mean of $1 - \delta_i$ over $i$ is defined as the overall modularity score.

### 8.6 F. IMPLEMENTATION DETAILS OF ANNOTATION PERTURBATION

To evaluate the effect of mislabeling phenotypic levels, we made use of the dataset of T4/T5 neurons, and generated three kinds of perturbation to the original labels:

First, we switched the phenotype labels of T4a neurons with one of the seven other types (T4b, T4c, T4d, T5a, T5b, T5c, T5d). In this scenario, phenotype labels of two cell types were incorrect, but the number of cell type clusters was the same. We had two levels of the dendritic phenotypes (T4/T5), and four levels of the axonal phenotypes (a/b/c/d). Therefore we kept one dimension for the dendritic feature, and three dimensions for the axonal feature.

Second, we merged the axonal phenotypic level a with another level (b/c/d), as an incorrect new level (a+b/a+c/a+d). In this scenario, we had three axonal phenotypes, therefore we kept two dimensions for the axonal feature.

Third, we randomly split each of the four axonal lamination labels (a/b/c/d) into two levels. For instance, among neurons with the original axonal level a, some of them were labeled with a level a1, and the others were labeled with a level a2. In this scenario, we had eight axonal phenotypes (a1/a2/b1/b2/c1/c2/d1/d2), and we kept seven dimensions for the axonal feature.

We performed FLDA on the dataset of T4/T5 neurons but with these perturbed annotations. Metrics from each of the perturbed annotations were measured and compared with those from the original annotation.

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

# 9 ADDITIONAL INFORMATION

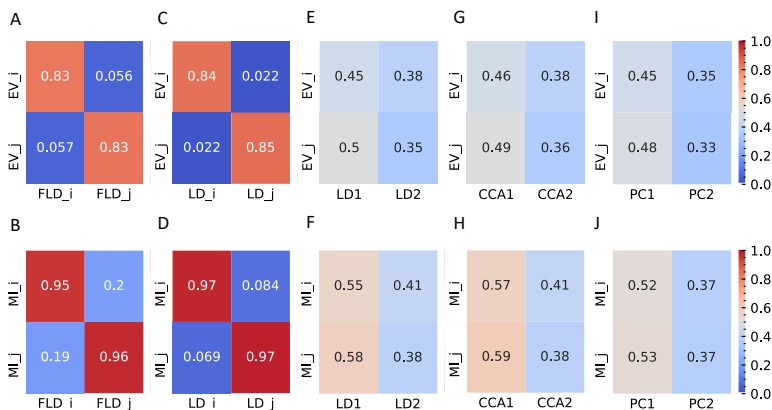

Figure 5: Representative plots (at $\sigma = 0.6$) of EV and MI metrics for FLDA and other models. (A,B) EV (A) and MI (B) metrics of FLDA. $FLD_i$ and $FLD_j$ indicate the factorized linear discriminants for features $i$ and $j$. (C,D) EV (C) and MI (D) metrics of 2LDAs. $LD_i$ and $LD_j$ indicate the linear discriminant components for features $i$ and $j$. (E,F) EV (E) and MI (F) metrics of LDA. $LD_1$ and $LD_2$ indicate the first two linear discriminant components. (G,H) EV (G) and MI (H) metrics of CCA. $CCA_1$ and $CCA_2$ indicate the first two canonical correlation axes. (I,J) EV (I) and MI (J) metrics of PCA. $PC_1$ and $PC_2$ indicate the first two principal components. $EV_i$ and $EV_j$ are the explained variance of features $i$ and $j$ along an axis, and $MI_i$ and $MI_j$ indicate the mutual inform between an axis and features $i$ and $j$ respectively. Values of EV and MI metrics are also indicated by the color bars on the right side.

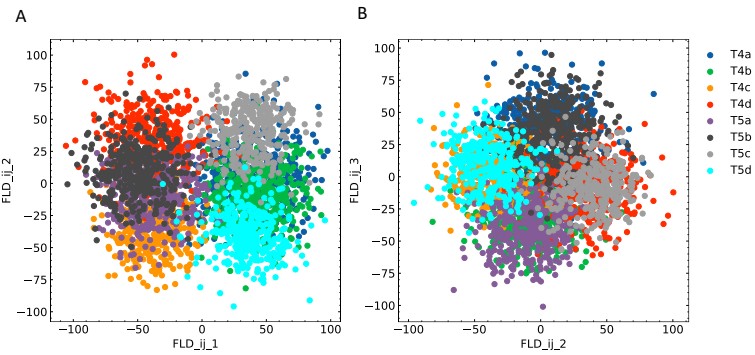

Figure 6: Additional plots for FLDA on the dataset of T4/T5 neurons. (A,B) Projection of the original gene expression data into the two-dimensional space made of the first and second ($\text{FLD}_{ij1}$ and $\text{FLD}_{ij2}$) (A) or the second and third ($\text{FLD}_{ij2}$ and $\text{FLD}_{ij3}$) (B) discriminant axes for the component that depends on the combination of both features $i$ and $j$. Different cell types are indicated in different colors as in (B).

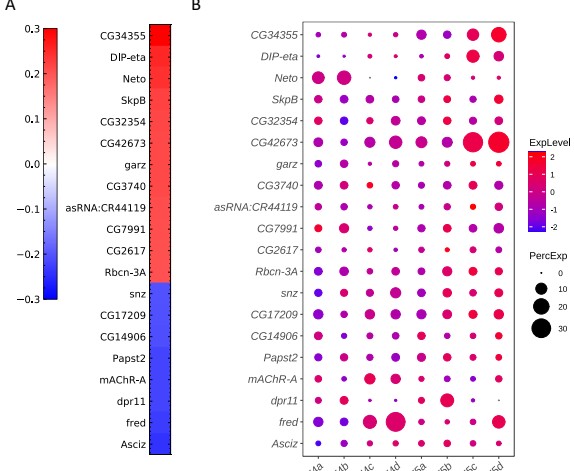

Figure 7: Additional plots for critical genes extracted from the sparse algorithm. (A) Weight vector of the 20 genes selected for the interaction of both dendritic and axonal features ($\text{FLD}_{ij1}$). The weight value is indicated in the color bar with color indicating direction (red: positive and green: negative) and saturation indicating magnitude. (B) Expression patterns of the 20 genes from (A) in eight types of T4/T5 neurons. Dot size indicates the percentage of cells in which the gene was expressed, and color represents average scaled expression.

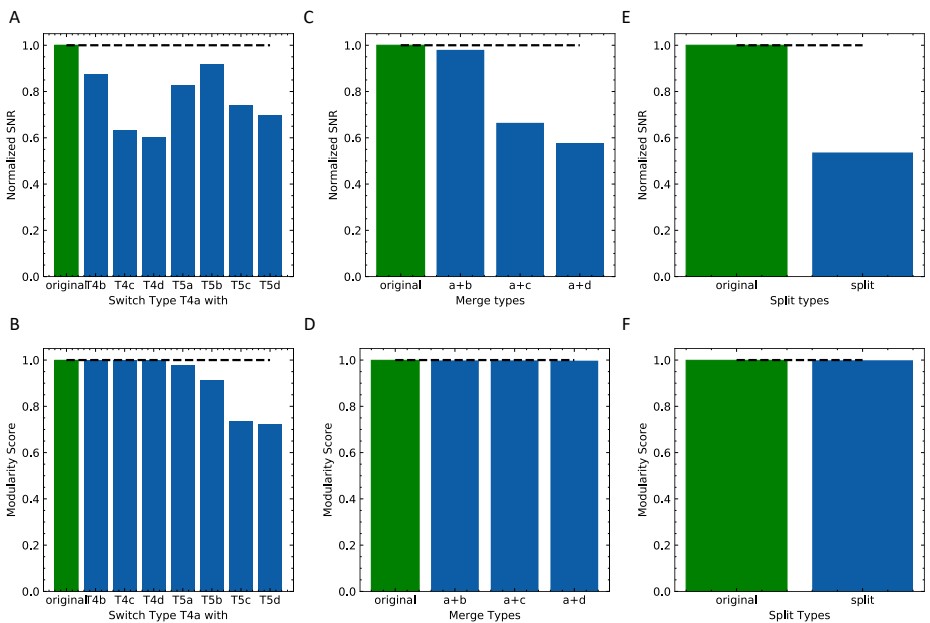

Figure 8: Evaluation of the effect of incorrect phenotype annotation on the dataset of T4/T5 neurons. (A,B) Normalized overall SNR metric (A) and overall modularity score (B) of FLDA after switching labels of T4a type with another neuronal type. (C,D) Normalized overall SNR metric (C) and overall modularity score (D) of FLDA after merging the axonal phenotypic level a with another phenotypic level (b/c/d). (E,F) Normalized overall SNR metric (E) and overall modularity score (F) of FLDA after splitting each axonal phenotypic level into two. Metrics under the original annotation are colored in green, and the values are indicated by the dashed lines. Here the SNR values are normalized with respect to that of the original annotation.

