# OpenReview forum: "Factorized linear discriminant analysis for phenotype-guided representation learning of neuronal gene expression data"
_ICLR.cc/2021/Conference — Reject_

### Official Review · AnonReviewer3 · 2020-10-28
**An interesting real world application**

**Rating:** 5
**Confidence:** 3

**Review:**

*Summary*
This manuscript presents a novel dimensionality reduction method, called Factorized Linear Discriminant Analysis. The method starts from a real problem in neurobiology, and tries to link expression levels of neural genes to phenotypes. In particular, the main goal of the proposed technique is to find linear projections of the genes expression which vary maximally with one phenotypical aspect and minimally with the others. The approach is evaluated using a synthetic example and a real case study involving Drosophila T4/T5 cells.


*Positive points:*
-> The paper starts from a real and challenging bioinformatics problem, i.e. the analysis of single-cell RNA sequencing data for neurobiology; data analysis tools for these data are nowadays fundamental to unravel the high complexity of this bio-medical field.
-> The idea is interesting, well motivated and well explained.
-> The proposed approach is simple and understandable: interpretability of solutions and results is currently fundamental when dealing with bio-medical data.
-> The method is tested using a real world application


*Negative points / questions:*

COMMENT 1.
The main problem of this manuscript is that the proposed method is not well inserted into the state of the art. Actually the discussion of authors of the related works is very limited, with only one paragraph at the bottom of page 1 plus the description of Linear Discriminant Analysis in Section 4. Many different linear dimensionality reduction techniques have been proposed in the past, each one with different characteristics, goals and optimization techniques. Authors should discuss them, especially in relation with their approach. A good entry point is the survey of Cunningham and Ghahramani:

John P. Cunningham, Zoubin Ghahramani: Linear Dimensionality Reduction: Survey, Insights, and Generalizations, Journal of Machine Learning Research 16 (2015) 2859-2900

Also for what strictly concerns Linear discriminant Analysis, I think that a deeper discussion is needed. Many different extensions of LDA have been proposed, some of them strictly related to the goals/methods of this paper, like 2D-LDA:

Ming Li, Baozong Yuan, 2D-LDA: A statistical linear discriminant analysis for image matrix,
Pattern Recognition Letters, Volume 26, Issue 5, 2005, Pages 527-532,

This contextualization with respect to the state of the art is fundamental: without this, it is very difficult to get the true contribution of the proposed approach. In the same spirit, I suggest the authors to include some more recent techniques in the experimental comparison.



COMMENT 2.
The sparisification approach has not been fully discussed and justified. Many different methods for sparsification have been proposed, why do authors choose this particular one? How does this method relate to alternatives? Some sparse algorithms have been introduced also for discriminant analysis (not strictly related to the medical field), such as:

N. H. Ly, Q. Du and J. E. Fowler, "Sparse Graph-Based Discriminant Analysis for Hyperspectral Imagery," in IEEE Transactions on Geoscience and Remote Sensing, vol. 52, no. 7, pp. 3872-3884, July 2014,


COMMENT 3 .
If I correctly understand, in the experiments authors apply a PCA to reduce the gene expressions before applying the proposed method (last two lines of page 5). Is this a reasonable choice? I know that this is commonly done in other scenarios (like in Fisher-faces for face recognition), but in these experiments genes are fundamental for the knowledge extraction. I guess this is not done for the sparsified version (otherwise genes would have not been extracted), can you comment on this?




COMMENT 4.
Some authors argued that formulating the optimization of linear dimensionality reduction techniques as eigenvalue or generalized eigenvalue problems is not always an adequate choice:

John P. Cunningham, Zoubin Ghahramani: Linear Dimensionality Reduction: Survey, Insights, and Generalizations, Journal of Machine Learning Research 16 (2015) 2859-2900

In such paper the authors also suggested an alternative. Can you provide a comment on this?


COMMENT  5.
I found some difficulties in reading and understanding the presentation and the discussion of the results. One problem was definitely the fact that tables are put in the appendix, and that such tables contains many numbers. I suggest the authors to put a summarizing table inside the manuscript (if possible).


COMMENT 6.
The phenotypical features are assumed to be categorical. How strict is this assumption? How much information are we loosing in this context? And in other contexts? In other words, are there other scenarios/applications in which this method be applied?  Adding some other possible application scenarios would increase the value of the proposal.


COMMENT 7.
This comment is related to the previous one, but contains more an open suggestion rather than a comment. Did you consider to use non categorical features? I think that this would open the usage of approaches used in three-way data analysis. Even I’m not aware of dimensionality reduction techniques methods for three-way data, I think a relation exists; techniques to extract interesting relations between different directions of data have been proposed, especially in the context of Tri-clustering, see for example:

Henriques, R., Madeira, S.C.: Triclustering algorithms for three-dimensional data analysis: a comprehensive survey. ACM Comput. Surv. 51(5), 95 (2019)

I suggest the authors to take a look also at this field.

---

> ### Author Response · Authors · 2020-11-20
> **Author responses**
>
> Response to COMMENT1:
>
> This is a valid criticism and we thank you for the suggested papers. We added several paragraphs in the Related Work section about the relation to prior work on linear dimensionality reduction.
>
> In terms of the "2D-LDA" paper, we think it is an interesting one, but it solves a somewhat different problem, where the input data of each instance are 2D (for example, a 2D image with m x n pixels), but with one supervision label (for example, one of n classes). In our problem, the data of each cell is 1D (g genes), but the supervision labels are a multi-dimensional Cartesian product (for example, a x b phenotypic features). We believe the “2LDAs” approach mentioned in the article is the closest baseline model for our problem.
>
> In general, we agree and we appreciate the reviewer’s suggestion. In addition to the comparison to other methods in linear dimensionality reduction, we also provide new results in the revised manuscript comparing our approach to PCA and CCA.
>
>
> Response to COMMENT2:
>
> We appreciate the reviewer’s comment. As detailed in the response to Reviewer 1, Rifle specifically addresses unique challenges for sparse generalized eigenvalue problems. In addition, Rifle is guaranteed to converge linearly to the solution with the optimal statistical rate of convergence, and its computational complexity scales linearly with the dimensionality of the data. The revised manuscript offers more justification and technical details of the approach.
>
> The Ly et al paper presents an interesting approach to obtain a sparse graph representation of the data (represented by a sparse matrix of edge weights between data points), and then perform dimensionality reduction using discriminant analysis. However, for the present data set we are interested in a specific form of sparse representation that has biological meaning, namely constraining the number of non-zero gene expressions. If applied to a different scientific domain, one can certainly imagine pairing FLDA with a different sparsification method.
>
>
> Response to COMMENT3:
>
> Yes, PCA preprocessing was not done for the sparsified version, and the revised manuscript makes that clear. In the non-sparse mode of FLDA, we think that PCA is a reasonable preprocessing choice because:
>
> 1. There exist strong correlations among different genes in the data set. For example, genes that are regulated by the same transcription factor will tend to vary up and down together. PCA groups these genes together, and helps us identify and get rid of noise in later components.
>
> 2. PCA is commonly used for data preprocessing. As mentioned in John P. Cunningham, Zoubin Ghahramani: Linear Dimensionality Reduction: Survey, Insights, and Generalizations, Journal of Machine Learning Research 2015, “the most common approach to undercomplete ICA is to preprocess the mixed data X with PCA (e.g., Joho et al., 2000), reducing the data to r dimensions, and running a standard square ICA algorithm.”
>
>
> Response to COMMENT4:
>
> As mentioned in this paper, “It is critical to clarify the distinction between these two methods: the heuristic and orthogonal solutions are indeed optimal, but for different objectives, as discussion in Section 3.1.3”. We are fully aware of the difference between the heuristic solution (from an eigenvalue objective as in FLDA) and the alternative orthogonal one. We chose our objective because we don’t want to impose orthogonality of the resulting axes, and we prefer a simple closed-form solution. The alternative objective has no closed-form solutions, and a gradient-based solution may result in a local optimum.
>
>
> Response to COMMENT5:
>
> We appreciate the suggestion and we agree that it is hard to get information from the large tables. The revised version color-codes part of the table to highlight the key comparisons.
>
>
> Response to COMMENT6:
>
> The categorical assumption is well validated for the data set in question, where the phenotypes are unambiguously defined in terms of the anatomy of the neurons. Similarly in the vertebrate retina, decades of research have established the sorting of cell types by anatomy and physiology. In recent months there have been explosive developments in the area of human cell typing, leading to gene expression data sets with many categorical clusters. As the structure and function of each of those cell types become better understood, FLDA can serve as a way to organize these cell types into tabular or tensor structures, similar to the periodic table of elements. In general, we think that FLDA can be applied to data that are organized into clusters whose labels are formulated as a Cartesian product of multiple features, for example, the data of face images, which can be labeled jointly by the age, gender, and other features of a person. Of course, there are other occasions where the labels vary instead over a continuous range, and a different statistical model will be more appropriate.

---

> > ### Author Response · Authors · 2020-11-20
> > **Author responses (continued)**
> >
> > Response to COMMENT7:
> >
> > Thanks again for the comment. An approach was recently proposed to take non-categorical phenotype features, combine them with gene expression data, and perform cross-modal clustering to get cell types (Gala et al. A coupled autoencoder approach for multi-modal analysis of cell types. NeuIPS 2019). This approach requires the collection of both transcriptome and phenotypic data of individual cells simultaneously, which is usually done by PatchSeq, a difficult and low-throughput experiment. Therefore, we see our approach as a good complement to this existing one. Again for brain regions such as the mouse retina and the Drosophila primary visual system, where plenty of studies have revealed the categorical phenotype of each cell type, we can make use of this prior knowledge, and just identify the relationship between gene expressions and the known phenotypes. If the connection to phenotypes is not yet established, a multi-modal clustering approach might be very useful.

---

### Official Review · AnonReviewer2 · 2020-10-28
**FLDA is a promising method for scRNA-seq analysis, which misses some deeper evaluation yet.**

**Rating:** 6
**Confidence:** 4

**Review:**

## Summary
The paper provides and ANOVA-inspired method, called FLDA, for creating a low dimensional embedding of scRNA-seq data. Additionally they propose a sparsity based method to find gene signatures, which can be used for further biological validation. The authors extensively evaluated their method on a data set of real expression values in Drosophila neurons.
They compared FLDA to two simpler and similar approaches, namely linear discriminant analysis (LDA) and a more feature-aligned version 2LDA.
On all benchmarks and metrics the FLDA shows clear advantage over the other methods.
​
## Reasons for score
Overall, I vote for accepting. I can imagine several use cases of this approach, which looks like it is computational very feasible and easy to apply.
Additionally it is theoretically well founded on existing statistical methods.
However, my main concern is FLDAs dependency on correct feature annotation, which is not mentioned by the authors.
I hope authors can address my concern and the other cons in the rebuttal period.
​
## Pros
- The authors clearly justify their design principle of FLDA based on ANOVA.
- Benchmarks using different scores show clear advantage of the FLDA method.
- The gene signature found for the T4/T5 data set show known marker genes for neuronal development.
- Extension to the >2 feature case and other possible extensions are clearly outlined.
- Extensive appendix showing theoretical validation for most obstacles in scRNA-seq data.

## Cons
- No evaluation on how dependent the model is on correct feature annotation.
- No comparison to other methods for estimating gene signatures.
- Validation on just one biological/real data set.​

## Questions during rebuttal period
Please address and clarify the cons above.
Mainly what happens if one level is incorrectly defined, e.g. the ground truth are two or more levels for this singular level. Or in other words how robust is the method to the annotation.

---

> ### Author Response · Authors · 2020-11-20
> **Author responses**
>
> Response: We addressed these questions as follows:
>
> Regarding the dependence of the model on correct feature annotation: Because FLDA is a supervised approach, a set of correct annotations is definitely important to extract useful information. But we can make use of FLDA to raise a flag when there could be incorrect annotations. As detailed in the last paragraph in the Results section, we designed a perturbation analysis. If FLDA on the perturbed annotations produces lower metric scores, that raises the confidence that the original annotation is correct.
>
> Regarding other methods for estimating gene signatures: The most common approach to extract gene signatures is to use differential gene expression analysis (DGE): define an experimental sample and a control sample, then find individual genes whose expression differs significantly between the samples based on a parametric or non-parametric statistical test. The main difference of our approach compared with DGE is that we impose sparseness on the entire gene vector during optimization, therefore we consider correlations between gene expressions, while DGE considers only one gene at a time.
>
> Regarding validation of FLDA on other biological data: We have applied the same method to gene expression of cell types in the vertebrate retina, whose phenotypes can be jointly described by their morphological (shape) and physiological (function) properties. Please see the response to Reviewer 1.

---

> > ### Comment · AnonReviewer2 · 2020-11-23
> > **Reviewer's reply**
> >
> > First of all thanks for your response.
> >
> >     Regarding the dependence of the model on correct feature annotation: Because FLDA is a supervised approach, a set of correct annotations is definitely important to extract useful information. But we can make use of FLDA to raise a flag when there could be incorrect annotations. As detailed in the last paragraph in the Results section, we designed a perturbation analysis. If FLDA on the perturbed annotations produces lower metric scores, that raises the confidence that the original annotation is correct.
> >
> >         Quote from the PDF:
> >
> >         To test this, we switched the phenotype labels of T4a neurons with one of the other types; second, a singular phenotypic level was incorrectly split into two or more. For this, we merged an axonal phenotypic level (a) with another (b/c/d); third, two or more phenotypic levels are incorrectly merged into one. To check this, we randomly split each of the four axonal lamination levels (a/c/b/d) into two levels.
> >
> > First: I don't understand why switching labels inside one location type should have any impact.
> > Second/Third. I think the explanations were confused in the text.
> > No effect of splitting times. Does this imply that I can split into as many classes as I want? At some point this should impair further analysis. Please discuss.
> >
> >     Regarding other methods for estimating gene signatures: The most common approach to extract gene signatures is to use differential gene expression analysis (DGE): define an experimental sample and a control sample, then find individual genes whose expression differs significantly between the samples based on a parametric or non-parametric statistical test. The main difference of our approach compared with DGE is that we impose sparseness on the entire gene vector during optimization, therefore we consider correlations between gene expressions, while DGE considers only one gene at a time.
> >
> > Thanks for the explanations and I see your point there.
> > Maybe I was a bit weak in explaining this cons, but I think an experimental comparison to another method for estimating gene signatures on your data set would be nice, e.g. do you find the same signatures with your method on the T4/T5 data set as other methods, and if they differ to what extent. Is it biologically meaningful?
> >
> >     Regarding validation of FLDA on other biological data: We have applied the same method to gene expression of cell types in the vertebrate retina, whose phenotypes can be jointly described by their morphological (shape) and physiological (function) properties. Please see the response to Reviewer 1.
> >
> > Thanks, I would like to see that draft or just another standard data set, where your method can be applied, e.g. GSE96583 (Sample 2.1/ 2.2).

---

> > > ### Author Response · Authors · 2020-11-24
> > > **Author responses**
> > >
> > > COMMENT1: "First: I don't understand why switching labels inside one location type should have any impact."
> > >
> > > Response: Thanks for the comment. The purpose here is to switch labels of the type T4a with another type, for example, T5a, and see how that affects the metric scores of FLDA. For the reference condition with unperturbed phenotype labels, our plot used a confusing legend “T4a is switched with T4a”. By definition, this has no impact. To eliminate this confusion we changed the legend to “original”.
> > >
> > > COMMENT2: "Second/Third. I think the explanations were confused in the text. No effect of splitting times. Does this imply that I can split into as many classes as I want? At some point this should impair further analysis. Please discuss."
> > >
> > > Response: Thanks for the comment. The reason that there was no effect of splitting is that we kept the same number of dimensions as the original annotation. Randomly splitting clusters introduces additional dimensions in the analysis, but variance along these axes is driven by within-cluster noise. These axes have lower SNR scores and therefore will not be selected if we keep the top n dimensions as the original annotation. However, this comment makes us realize that we should change the number of selected dimensions based on the number of levels in the new annotations. With that, we run the perturbation analysis and show that splitting decreases the metric score of SNR. We expand on this perturbation analysis and included paragraphs in Appendix F to make it clear.
> > >
> > > COMMENT3: "Thanks for the explanations and I see your point there. Maybe I was a bit weak in explaining this cons, but I think an experimental comparison to another method for estimating gene signatures on your data set would be nice, e.g. do you find the same signatures with your method on the T4/T5 data set as other methods, and if they differ to what extent. Is it biologically meaningful?"
> > >
> > > Response: Thanks for the comment. Indeed, in the original paper where the dataset of T4/T5 neurons was reported (Kurmangaliyev et al. Modular transcriptional programs separately define axon and dendrite connectivity. Elife. 2019), the group performed their own analysis and extracted gene signatures using conventional methods. See their Figures 2-4 and the summary Figure 7. Our results largely overlap with theirs, and confirm the indicator genes they tested in the experiment, for example, genes determining T4 vs T5 neurons, such as “TfAP-2”, “dpr2”, and genes determine a/b vs c/d types, such as “bi”, “klg”, and “pros”. We did find additional genes such as “pHCl-1” for the dendritic T4 vs T5 types and “Lac” for the axonal a/b vs c/d types. These genes make sense to our knowledge. For example, “pHCl-1” is a chloride channel, potentially important for dendritic synaptic integration. “Lac” is a cell surface protein that belongs to the Ig superfamily, critical for cell morphogenesis. We also state this comparison more explicitly in the revised manuscript.
> > >
> > > COMMENT4: "Thanks, I would like to see that draft or just another standard data set, where your method can be applied, e.g. GSE96583 (Sample 2.1/ 2.2)."
> > >
> > > Response: We will provide a link in the private channel for the draft where we applied FLDA to other datasets. Please note that we only include relevant paragraphs to show validation of FLDA on other biological data and we remove author information to keep anonymity. There are also recent papers from Allen Institute of Brain Science where datasets about morphology, physiology, and transcriptomes of neurons will be available in short order and are good candidates for this analysis (Bakken et al. Single-cell RNA-seq uncovers shared and distinct axes of variation in dorsal LGN neurons in mice, non-human primates and humans. Biorxiv. 2020; Gouwens et al. Integrated morphoelectric and transcriptomic classification of cortical GABAergic cells. Cell. 2020).

---

### Official Review · AnonReviewer4 · 2020-10-29
**This paper, which describes a straightforward linear factorization method to try to explain single-cell RNA-seq data, solves a niche problem**

**Rating:** 5
**Confidence:** 2

**Review:**

This manuscript describes a generalization of ANOVA that is intended to be used in the interpretation of single-cell RNA-seq data. The method requires specification of an orthogonal, discrete categorization of cells, nominally by phenotype.  The method then linearly factorizes the observed gene expression values into features and their interactions, relative to the phenotypic categories.  The factors can then be used to help interpret the categories, especially in conjunction with a regularizer to reduce the number of genes involved in the factors.

I found this paper frustrating to read.  The second paragraph does not make clear exactly what problem is being addressed.  It says that we are "given phenotypic descriptions of neuronal types," but not where those descriptions come from.  So I skipped ahead to the methods section to try to figure it out.  But even after reading that section, I could not understand where the phenotype values come from.  It was only when I made it to Section 2 that I verified that, indeed, the phenotypes are not observed but only inferred.

Having understood the problem, it seems to me that the use case for this approach is quite specific.  We need to have RNA-seq data that can be clustered in such a way that we can assign clusters to pre-defined phenotypic categories.  The claim, in the discussion section, that "our approach can be easily generalized to .. additional characteristics such as electrophysiology and connectivity" was not clear to me, but I assume this still refers to phenotypes that are inferred from the scRNA-seq data.

The critique leveled against CCA is that "this approach cannot factorize gene expressions according to individual features, making the result hard to interpret."  But this seems like it must also be true of the proposed method, since prior to any analysis the data is transformed via PCA (Section 5.2).  I am confused, therefore, about how the method can be used to select genes in Section 6.

The comparison to LDA is done using two metrics, based on signal-to-noise ratio and mutual information.  I would have liked to hear more about why these particular metrics are appropriate, and in particular how they relate to whatever use case the authors have in mind. Overall, I am still not convinced that this is a problem that needs to be solved.

---

> ### Author Response · Authors · 2020-11-20
> **Author responses**
>
> COMMENT1: "I found this paper frustrating to read. The second paragraph does not make clear exactly what problem is being addressed. It says that we are "given phenotypic descriptions of neuronal types," but not where those descriptions come from. So I skipped ahead to the methods section to try to figure it out. But even after reading that section, I could not understand where the phenotype values come from. It was only when I made it to Section 2 that I verified that, indeed, the phenotypes are not observed but only inferred."
>
> RESPONSE: We appreciate the concern about circular logic here, but “only inferred” is not quite correct. There is a handful of marker genes whose expression uniquely identifies each of the 8 phenotypes in the data set. These relationships are well established from decades of prior biological research. Those few marker genes are used to attach the phenotypic labels to the clusters in gene expression space. Our FLDA method then leverages those phenotypic labels to interpret structure among the many thousands of other genes, whose relation to the phenotypes was previously unknown.
>
> COMMENT2: "Having understood the problem, it seems to me that the use case for this approach is quite specific. We need to have RNA-seq data that can be clustered in such a way that we can assign clusters to pre-defined phenotypic categories (that’s the case here…). The claim, in the discussion section, that "our approach can be easily generalized to .. additional characteristics such as electrophysiology and connectivity" was not clear to me, but I assume this still refers to phenotypes that are inferred from the scRNA-seq data."
>
> RESPONSE: The method of marker genes is by far not the only option for labeling transcriptomic data. For example cells can be isolated one by one from the tissue based on certain phenotypic characters and then sequenced in separate tubes. One can even measure a neuron’s electrical properties and photograph its shape before drawing out the cellular contents for sequencing. In still other methods, the sequencing is done in situ with the cell still in the tissue, allowing an assessment of its anatomy and other features. All these approaches to cell typing are developing at a dizzying pace. Our FLDA method applies to all these cases.
>
> COMMENT3: "The critique leveled against CCA is that "this approach cannot factorize gene expressions according to individual features, making the result hard to interpret." But this seems like it must also be true of the proposed method, since prior to any analysis the data is transformed via PCA (Section 5.2). I am confused, therefore, about how the method can be used to select genes in Section 6."
>
> RESPONSE: As detailed in the response to Comment 3 of Reviewer 3, we think that PCA is a reasonable preprocessing choice because PCA is commonly used for data preprocessing and there exist correlations in gene expressions. The problem of CCA is not the linear transformation of gene expressions, but the linear combination of phenotypic features. We re-organized the discussion of CCA, moved it into the section on Related Work, and expanded it with additional mathematical details. In addition, we provide results comparing our approach to CCA and the difference is easy to see. Finally, we confirm that PCA preprocessing was not done for the sparsified version of FLDA, and we make it clear in the revised manuscript.
>
> COMMENT4: "The comparison to LDA is done using two metrics, based on signal-to-noise ratio and mutual information. I would have liked to hear more about why these particular metrics are appropriate, and in particular how they relate to whatever use case the authors have in mind. Overall, I am still not convinced that this is a problem that needs to be solved."
>
> RESPONSE: As mentioned in the paper, there are two requirements for the low-dimensional embedding space: First, we want to recover cell types in the low-dimensional embedding. This requires that cells of the same type lie close to each other in the embedding space, but cells of different types lie far apart. This relative separation of cell types is measured by the signal-to-noise metric. Second, for the purpose of biological understanding, we seek axes in the low-dimensional space that are aligned with phenotypic features. This alignment can be measured and quantified using mutual information and its derived metric, such as the modularity score. In this way, we think the metrics used are directly relevant to the problem we want to solve. This is explained better in the revised manuscript.

---

### Official Review · AnonReviewer1 · 2020-10-30
**The paper presents a nice method in gene expression data analysis which has scientific implications, but probably lacks significance from a machine learning perspective**

**Rating:** 5
**Confidence:** 3

**Review:**

Pros:
The paper studies a very important problem in gene data analysis. The proposed method is technically sound. The method is intuitive in its idea and easy to implement. The results are interpretable. And according to the experimental evaluations, the proposed method is consistent to existing biological observations and could further identify unknown genetic targets. Therefore, it, potentially, has insightful scientific implications.

Cons:
- Relevance and Generalizability are unclear: The paper is relevant to researchers in subareas only and it is best-suited to a bioinformatics or neurobiology community. The paper requires background in neurobiological data analysis to evaluate whether the proposed method brings meaningful scientific insights.  In terms of novelties in machine learning field, the improvement over existing works, algorithmically or computationally, seems relatively limited. It is also unclear whether the proposed method generalizes well outside the subdomain of gene expression. It seems FLDA could possibly be applied to any such tensor data but a discussion on its general applicability to other data will be nice.

- Relation to prior work is insufficient: Related work has not been discussed adequately, thus making the significance of this paper unclear. The authors discusses CCA and autoencoders in introduction, and also mentions the differences between FLDA and LDA (2LDAs), which are compared with the proposed method in experiments. I find such discussions are inadequate for the readers to understand the baseline or state of the art in this field. Therefore, It is somewhat unclear how the work improves from existing works.

- Experimental evaluation is incomprehensive: As a largely application work, the comprehensiveness or tricks in experiments should be explained more clearly. For example, one part of the experiment, the sparsity-based regularization of FLDA seems like the application of Rifle (Kean Ming Tan et al) on the data. Readers are unclear what are the unique challenges in current setting. Also, any computational details (convergence, scalability, etc.) or any guidance on the hyperparameter selection? And does this method also generate meaningful results on another gene data?

---

> ### Author Response · Authors · 2020-11-20
> **Author responses**
>
> Comment1: Relevance and Generalizability are unclear.
>
> Response: We appreciate these concerns but respectfully disagree with the reviewer on this point. FLDA was developed to address the problem of finding a low-dimensional representation of high-dimensional gene expression data that is labeled by certain phenotypic features. As Reviewer 3 kindly pointed out, this is a novel method for linear dimensionality reduction, which is an important subfield of machine learning and a cornerstone for data analysis. FLDA can be applied to data other than gene expressions, as long as the data are organized into clusters whose labels are formulated as a Cartesian product of multiple features. For example, this might include face images, which can be labeled jointly by the age, gender, and other features of a person.
>
> Comment2: Relation to prior work is insufficient.
>
> Response: We agree and added several paragraphs in the Related Work section about the relation to prior work on linear dimensionality reduction. The revised manuscript also provides additional results comparing the approach to PCA and CCA.
>
> Comment3: Experimental evaluation is incomprehensive.
>
> Response: As the reviewer suggested we provided additional information and explanation:
> 1. Unique challenges in the current setting: “This is known as a sparse generalized eigenvalue problem, which has three unique challenges, as listed in \citet{tanSparseGeneralizedEigenvalue2018}: first, when the data are very high-dimensional, $\mM_{e}$ can be singular and non-invertible; second, because of the normalization term in the denominator, many solutions for sparse eigenvalue problems cannot be applied directly; finally, this problem involves maximizing a convex objective over a nonconvex set, which is NP-hard.”
> 2. Computational details and hyperparameter selection of Rifle: “As proved in \citet{tanSparseGeneralizedEigenvalue2018}, if there is a unique sparse leading generalized eigenvector, Rifle will converge linearly to it with the optimal statistical rate of convergence. The computational complexity of the second step is $O(lg+g)$ for each iteration, therefore Rifle scales linearly with $g$, the dimensionality of the original data. Based on the theoretical proof, to guarantee convergence, the hyperparameter $\eta$ was selected to be sufficiently small such that $\eta \lambda_{max}(\mM_{e}) < 1$, where $\lambda_{max}(\mM_{e})$ is the largest eigenvalue of $\mM_{e}$. In our case, the other hyperparameter $l$, indicating how many genes to be preserved, was empirically selected based on the design of a follow-up experiment. As mentioned later in Results, we chose $l$ to be 20, a reasonable number of candidate genes to be tested in a biological study.”
>
> We did apply FLDA to other gene expression data. One example is the gene expression of cell types in the vertebrate retina, whose phenotypes can be jointly described by their morphological (shape) and physiological (function) properties. FLDA not only extracted gene signatures consistent with previous reports but also suggested new gene candidates for certain phenotypes. With a team of collaborators, we are working on another biology-oriented article about these retinal neuronal types. If necessary, we can provide a link to a draft in which the results are described.

---

### Author Response · Authors · 2020-11-20
**Author responses**

Response: We thank all the reviewers for their insightful comments! We addressed all the reviewers’ questions below. Based on the feedback, we provided additional context and information in the revised manuscript:
1. Challenges of the sparse generalized eigenvalue problem and technical details of Rifle, the sparse algorithm.
2. Background of linear dimensionality reduction and its state-of-art alternative models.
3. New results comparing FLDA with PCA and CCA.
4. Results of a new perturbation analysis to flag possibly incorrect annotation.
5. Discussion on how FLDA can generalize to data other than gene expressions.

---

### Decision · Program_Chairs · 2021-01-07
**Final Decision**

**Decision:**

Reject

**Comment:**

The paper introduces a linear projection method, inspired by ANOVA,
for finding a supervised low-dimensional embedding.

A positive aspect is that the method is straightforward, and it is
even slightly surprising that in the family of linear models, there
still was an uncovered "niche".

The paper was considered useful for the purpose studied in the
paper, single-cell RNA-seq data analysis. But to claim broader
usefulness, more evidence should be presented.

One particular detail which was brought up by all reviewers was the
PCA preprocessing. For ICA it is a sensible choice, as linear ICA is
essentially "just" a rotation of the PCA components. But the
justification is not as good for a supervised method. PCA may be
necessary in practice, but may lose important category-relevant
information.

The paper still needs a significant revision before publication.  Even
though the method is straightforward method, a lot of time and
discussion was required for expert reviewers to understand it.